# Cycloartocarpin Inhibits Migration through the Suppression of Epithelial-to-Mesenchymal Transition and FAK/AKT Signaling in Non-Small-Cell Lung Cancer Cells

**DOI:** 10.3390/molecules27238121

**Published:** 2022-11-22

**Authors:** Sucharat Tungsukruthai, Boonchoo Sritularak, Pithi Chanvorachote

**Affiliations:** 1Division of Health and Applied Sciences, Faculty of Science, Prince of Songkla University, Hat Yai 90110, Thailand; 2Department of Pharmacognosy and Pharmaceutical Botany, Faculty of Pharmaceutical Sciences, Chulalongkorn University, Bangkok 10330, Thailand; 3Center of Excellence in Cancer Cell and Molecular Biology, Faculty of Pharmaceutical Sciences, Chulalongkorn University, Bangkok 10330, Thailand; 4Department of Pharmacology and Physiology, Faculty of Pharmaceutical Sciences, Chulalongkorn University, Bangkok 10330, Thailand

**Keywords:** cycloartocarpin, migration, metastasis, epithelial–mesenchymal transition (EMT), lung cancer

## Abstract

Lung cancer metastasis is a multifaceted process that accounts for 90% of cancer deaths. According to several studies, the epithelial–mesenchymal transition (EMT) plays an essential role in lung cancer metastasis. Therefore, this study aimed to investigate the potential pharmacological effect of cycloartocarpin on the suppression of metastasis-related behaviors and EMT. An MTT assay was used to examine cell viability. Cell migration was determined using a wound healing assay. Anchorage-independent cell growth was also performed. Western blot analysis was used to identify the key signaling proteins involved in the regulation of EMT and migration. The results found that non-toxic concentrations of cycloartocarpin (10–20 μM) effectively suppressed cell migration and attenuated anchorage-independent growth in H292, A549, and H460 cells. Interestingly, these effects were consistent with the findings of Western blot analysis, which revealed that the level of phosphorylated focal adhesion kinase (p-FAK), phosphorylated ATP-dependent tyrosine kinase (p-AKT), and cell division cycle 42 (Cdc42) were significantly reduced, resulting in the inhibition of the EMT process, as evidenced by decreased N-cadherin, vimentin, and slug expression. Taken together, the results suggest that cycloartocarpin inhibits EMT by suppressing the FAK/AKT signaling pathway, which is involved in Cdc42 attenuation. Our findings demonstrated that cycloartocarpin has antimetastatic potential for further research and development in lung cancer therapy.

## 1. Introduction

Lung cancer represents one of the most common malignant tumors that potentially threaten human life due to its high incidence and mortality [1]. The majority of lung cancer deaths—nearly 80–90%—are due to malignant metastasis. In fact, when metastases have already formed in lung cancer patients, who are frequently diagnosed at an advanced stage, targeted therapy is very challenging and systemic therapy is less successful [2,3].

According to the metastasis process, cancer cells can invade and spread from the primary tumor to a wide range of organs through the blood and lymphatic systems. Additionally, a variety of mechanisms must be used by cancer cells to acquire their migratory and invasive characteristics. One of the most significant pathways is epithelial-to-mesenchymal transition (EMT). The epithelial cells enhance their motility regulatory signals during EMT, which induces motile and invasive phenotypes and allows them to survive in anchorage-independent circumstances [4,5].

During the EMT process, epithelial cancer cells become mesenchymal cells that are ready to migrate by expressing mesenchymal proteins such as vimentin, N-cadherin, slug, and snail. In contrast, the adhesive molecules such as E-cadherin are depleted, which increases cancer aggressiveness [6]. Previous research found that vimentin, a crucial EMT marker, is overexpressed in epithelial cancers such as lung, prostate, and breast cancer, while high expression levels are linked to poor prognosis [7]. Furthermore, EMT is also influenced by several transcription factors, including snail and slug. Significantly, snail knockdown altered EMT processes while also suppressing tumor cell proliferation and invasion in lung cancer cells [8].

There is mounting evidence that ATP-dependent tyrosine kinase (AKT) and focal adhesion kinase (FAK) play significant roles in lung cancer metastasis [9,10]. According to the mechanistic approach, the activation of FAK regulates the signal via the phosphorylation of AKT, resulting in cell migration in various cancers [11,12]. Additionally, the highly metastatic phenotypes are promoted by activation of the FAK and AKT/mammalian target of rapamycin (mTOR) pathways [13]. It was also discovered that the Rho family, especially cell division cycle 42 (Cdc42), is upregulated by phosphorylated AKT (p-AKT), which has been shown to be important in cell migration and actin rearrangement [14,15]. Therefore, the compounds with the potential to inhibit migration by focusing on these signaling pathways are of interest as promising candidates for lung cancer treatment and cancer prevention.

Cycloartocarpin is a natural compound extracted from *Artocarpus gomezianus* (Moraceae). According to reports, cycloartocarpin has an antibacterial effect, tyrosinase inhibitory activity, and a DPPH free radical scavenging ability. In addition, it was also discovered in *Artocarpus heterophyllus Lam*. (Moraceae) heartwoods [16,17]. However, the effect of cycloartocarpin on the metastasis and EMT of lung cancer cells, as well as its molecular mechanisms, have not yet been completely clarified. Consequently, the objective of this study was to examine the effect of cycloartocarpin on cancer migration and EMT in human lung cancer cells. The results of this study could support the development of cycloartocarpin as a novel antimetastatic treatment for lung cancer.

## 2. Results

### 2.1. Cytotoxic Effect of Cycloartocarpin on Human Non-Small-Cell Lung Cancer Cells

According to the aforementioned study, cycloartocarpin (Figure 1A) has an antibacterial effect and the capacity to scavenge DPPH free radicals [17]. However, there is no scientific evidence that cycloartocarpin has antimetastatic activity. This information might be helpful in the development of cycloartocarpin for further clinical research. First, we identified the non-toxic concentrations of cycloartocarpin to be used in the subsequent experiments. We also determined the cytotoxic effect of cycloartocarpin in other cell lines, including a non-tumorigenic lung epithelial cell line (BEAS-2B) and human keratinocyte cell line (HaCaT), for comparison. Briefly, the cells were treated with cycloartocarpin (0–100 μM) for 24 h. The cell viability was evaluated by an MTT assay. The results showed that cycloartocarpin did not significantly reduce the viability of NSCLC cells at concentrations lower than 20 μM (Figure 1B–D). Moreover, the results showed that all normal cells were unaffected by cycloartocarpin at concentrations of 0 to 100 µM (Figure 1E,F). Furthermore, the half maximal inhibitory concentration (IC_50_) of cycloartocarpin was found to be 38.62 ± 0.35, 40.76 ± 0.46, and 43.18 ± 0.67 μM in H292, A549, and H460 cells, respectively (Figure 1G).

The process of cell death is typically divided into two mechanisms: apoptosis and necrosis, based on the various ways in which cell death is stimulated and results in biochemical and morphological changes in cells [18]. The nuclear staining and fluorescence microscopy techniques are most frequently employed to visualize the alteration in nuclear morphology that occurs in both apoptotic and necrotic cells.

Mechanistically, the direct DNA binding capabilities of the stains or dyes allow them to emit fluorescence that can specifically represent the manner of death. In healthy cells, the nuclei are generally spherical, and the DNA is regularly distributed. In contrast, during apoptosis the DNA becomes condensed and is labeled with Hoechst33342, which subsequently emits blue fluorescent light, but this process does not appear during necrosis. In addition, the PI staining of the swelling cells allows them to be identified as necrosis cells [19]. Therefore, to further confirm whether cycloartocarpin at concentrations from 0 to 20 μM could be regarded as non-cytotoxic to the lung cancer cells, the occurrence of apoptotic and necrotic cells was determined by Hoechst33342/PI nuclear staining.

Cells were treated for 24 h with cycloartocarpin at concentrations ranging from 0 to 100 μM. Following that, the cells were stained with Hoechst 33,342 and propidium iodide (PI). We discovered that cycloartocarpin at 0–20 μM had no significant effect on apoptosis or necrosis cell death in all NSCLC cells (Figure 1H–J). For the following experiments, non-toxic concentrations (0–20 μM) of cycloartocarpin were used.

### 2.2. Cycloartocarpin Suppressed Motility in Human Lung Cancer Cells

To find out how cycloartocarpin affects cell migration, a wound healing assay was performed. Briefly, the monolayer of lung cancer cells was scratched. Cells were exposed to non-toxic concentrations of cycloartocarpin (0–20 μM) for 24 and 48 h. The suppressive effect on the cell motility of H292 and H460 cells was shown by a wider wound space in the cells incubated with 10–20 μM of cycloartocarpin, whereas cycloartocarpin at 5 μM had no significant effect on cell migration at 24 and 48 h (Figure 2A,C). For A549 cells, cell migration was inhibited by cycloartocarpin at 10–20 μM for 24 and 48 h. In addition, cycloartocarpin at 5 μM also considerably reduces the migration of the A549 cells after 48 h (Figure 2B).

### 2.3. Cycloartocarpin Attenuates Anchorage-Independent Growth of Human Lung Cancer H460, H292, and A549 Cells

The anchorage-independent growth of cancer cells represents both the ability of malignant tumor cells to spread and the resistance to anoikis [20,21]. Thus, we examined how cycloartocarpin affected cell growth and survival in detachment circumstances. Cells were developed in soft agar and were treated with or without cycloartocarpin for 7, 14, and 21 days. In comparison to the untreated control group, the size and number of the growing cancer colonies were investigated and calculated. The results demonstrated that cells treated with cycloartocarpin at concentrations of 5 to 20 μM had a dramatically reduced ability to form colonies. The reduction in colony number and size when compared to the control treatment further suggested that treatment with the compound may inhibit cell growth in H292 (Figure 3A), A549 (Figure 3B), and H460 cells (Figure 3C). These findings demonstrated that cycloartocarpin could reduce the survival and growth of lung cancer cells in the detached condition.

### 2.4. Cycloartocarpin Suppresses EMT by Inhibiting FAK/AKT Signaling Pathway

EMT is a biological mechanism that has been closely linked to cancer progression and metastasis. It is also known to promote cancer cell motility, invasion, anoikis resistance, and cancer stem cells in lung cancer [6,20,21]. As a result, we first looked into the EMT makers in cells treated with cycloartocarpin to assess the underlying mechanism. Cycloartocarpin (0–20 μM) was used to treat H292 and A549 cells for 24 h. The levels of the well-known EMT markers, vimentin, N-cadherin, slug, and snail, were measured. According to the findings, vimentin, N-cadherin, and slug levels in H292 (Figure 4A–D) and A549 cells (Figure 4F–I) were all significantly reduced by cycloartocarpin in a concentration-dependent manner. However, we discovered a significant decrease in the snail protein only in H292 cells (Figure 4E). Collectively, these findings support the inhibitory effect of cycloartocarpin on EMT in human lung cancer cells.

Additionally, prior studies indicated that upstream pathways including p-FAK/FAK, p-AKT/AKT, and Cdc42 signals controlled the EMT process [22]. Thus, we further investigated the protein levels of these proteins in cycloartocarpin-treated human lung cancer cells. The results found that cycloartocarpin significantly reduced p-FAK/FAK protein levels at the concentrations of 5–20 μM in H292 and A549 cells (Figure 5A,B,E,F). Previous research found that FAK phosphorylation activated its kinase activity, which activates AKT signaling [23]. Therefore, we further examined the promising effect of cycloartocarpin on AKT activation. We demonstrated that the phosphorylation of AKT was significantly affected by cycloartocarpin in H292 and A549 cells (Figure 5C,G).

Moreover, Cdc42 has been linked to increased migration. Therefore, the protein level of Cdc42 was further investigated. We discovered that cycloartocarpin significantly diminished the amount of Cdc42 protein in H292 and A549 cells (Figure 5D,H). These findings suggested that cycloartocarpin suppressed cell migration as well as the EMT process via a FAK/AKT-dependent mechanism involving Cdc42 attenuation.

## 3. Discussion

Human lung cancer has been described as having aggressive characteristics due to high proliferation, resistance to cell death, and metastasis progression [24]. Most lung cancer patients have metastatic disease at the time of their diagnosis. Moreover, the 5-year survival rate of lung cancer with metastasis is approximately 8% [25]. Cancer metastasis, in fact, is a multifaceted process that includes the spread of cancer cells from a primary tumor to various sites throughout the body via blood and lymphatic systems [1]. In addition, cancer cells can develop the ability to migrate and invade tissue, prevent anoikis, survive in the vascular system, and create colonies at other locations during the metastatic process [26].

The metastatic process of human cancer cells depends on the epithelial-to-mesenchymal transition (EMT), a process that causes the loss of epithelial features and the acquisition of mesenchymal features like invasiveness, motility, and successful metastatic colonization, as well as resistance to apoptosis cell death [27]. The EMT markers are the increase in N-cadherin, slug, snail, and vimentin, and subsequently the downregulation of E-cadherin [6,28]. In addition, the EMT phenotype is a significant marker of poor prognosis in lung cancer [29]. Many studies reveal the potential advantages of preventing cancer cells from migrating and the EMT process as a novel approach for metastasis prevention resulting in improved clinical outcome [30,31].

Recently, it was shown that compounds isolated from *Artocarpus gomezianus*, the Thai medicinal plant, have promising antimetastatic activity in human lung cancer cells [32,33,34]. Here, we have revealed novel information about the effect of the new active compound, cycloartocarpin (Figure 1A), extracted from *A. gomezianus*, on EMT and cell migration. We found that cycloartocarpin suppresses cell migration, as shown in Figure 2. Furthermore, cycloartocarpin has the potential to suppress anchorage-independent survival, which is the first and most important step in cancer metastasis (Figure 3). Treatment of lung cancer cells with cycloartocarpin at non-toxic concentrations (0–20 μM) significantly diminished the expression levels of EMT markers, such as vimentin, N-cadherin, and Slug (Figure 4), demonstrating the effect of cycloartocarpin in terms of its inhibition of EMT. Previous studies have shown that natural-product-derived compounds have the potential to inhibit cancer cell migration and EMT. For example, epicatechin-3-gallate (ECG) could suppress mesenchymal markers and phosphorylate FAK in human lung cancer cells [35]. Moreover, arctigenin has been shown to reduce N-cadherin and snail expression and increase the expression of E-cadherin in a concentration- and time-dependent manner [36]. Ovalitenone also showed inhibitory effects on migration and EMT by inhibiting the AKT/mTOR signaling pathway [37]. Similarly, phoyunnanin-E suppressed migration, growth, and EMT proteins such as vimentin, N-cadherin, and snail, together with slug expression [38].

Cancer cell migration includes several mechanistic pathways such as FAK [11], AKT [39], and Cdc42 [15]. FAK signaling pathways can promote cell migration and invasion [23]. Similarly, in high-motility cancer cells, FAK phosphorylation at Tyr 397 is necessary [40]. Focusing on AKT protein, in migrating cells AKT was discovered to be highly activated. [41]. Moreover, AKT is recognized as a crucial signaling pathway that regulates cell motility, survival, and proliferation [42]. It has been demonstrated that activated AKT is essential for metastasis and the EMT process in cancer cells [43,44]. Consequently, numerous compounds have been shown to reduce metastasis by interfering with FAK/AKT expression. For instance, a previous study found that artonin E suppressed cell migration through suppressing the FAK/AKT signaling pathway [45]. Furthermore, excisanin A inhibits breast cancer cell migration and invasion by attenuating the integrin β1/FAK/PI3K/AKT/β-catenin signaling [46]. Moreover, kaempferol could suppress the metastasis of tumors by the downregulation of the AKT and FAK pathways [47]. In this investigation, we discovered that the levels of proteins, such as p-FAK and p-AKT, were significantly decreased when treated with non-toxic concentrations of cycloartocarpin in A549 and H292 cells (Figure 5). Cell migration is also inhibited by decreased Cdc42 expression [48]. In the current study, we discovered that treating lung cancer cells with cycloartocarpin resulted in a decrease in cellular Cdc42 levels (Figure 5). In summary, we discovered the novel effects of cycloartocarpin in the suppression of cell migration, anchorage-independent growth, EMT by inhibiting the FAK/AKT signaling pathway, and Cdc42 (Figure 6). These findings indicate that cycloartocarpin has a promising effect on human lung cancer cells, suggesting that it could be used as an antimetastatic compound in cancer treatment.

## 4. Materials and Methods

### 4.1. Cycloartocarpin Isolation

Cycloartocarpin was isolated from the roots of *Artocarpus gomezianus* (Moraceae) as previously described [16]. Briefly, dried powder of the roots of this plant were macerated with petroleum ether to give a petroleum ether extract (25 g). The petroleum ether extract was subjected to vacuum liquid chromatography (silica gel, acetone–petroleum ether, gradient) to give nine fractions (A–I). Fraction F (1.6 g) was separated by Sephadex LH-20 (MeOH) and then by preparative thin-layer chromatography (silica gel, EtOAc-hexane, 1:4). The target fraction was purified again by Sephadex LH-20 (acetone) to afford cycloartocarpin (34 mg) as a yellow powder.

### 4.2. Preparation of Cycloartocarpin Stock Solution

Cycloartocarpin 50 mM stock solutions were prepared by dissolving cycloartocarpin in dimethyl sulfoxide (DMSO) solution and storing at −20 °C. They were then diluted in culture medium to achieve the final concentrations (0–100 μM) before being used in the treatment. The final DMSO concentration was 0.5% solution, which showed no cytotoxicity.

### 4.3. Cell Cultures and Reagents

H460, H292, A549, BEAS-2B, and HaCat cells were obtained from the American Type Culture Collection (Manassas, VA, USA). A549, BEAS-2B, and HaCat cells were cultured in Dulbecco’s Modified Eagle’s Medium (DMEM) (Gibco, Grand Island, NY, USA), whereas H460 and H292 were cultured in Roswell Park Memorial Institute (RPMI) 1640 medium (Gibco, Grand Island, NY, USA). The media were supplemented with 2 mM L-glutamine (Gibco, Grand Island, NY, USA), 100 U/mL penicillin–streptomycin (Gibco, Grand Island, NY, USA), and 10% fetal bovine serum (FBS) (Merck, DA, Germany). Hoechst 33342, Propidium iodide (PI), 3-[4, 5-dimethylthiazol-2-yl]-2,5-diphenyltetrazolium bromide (MTT), and Triton X-100 were obtained from Sigma Chemical, Inc. (St. Louis, MO, USA). Agarose was obtained from Bio-Rad Laboratories (Hercules, CA, USA). Bovine serum albumin (BSA) was obtained from Merck (DA, Germany). Antibodies for Vimentin (#5741), N-cadherin (#13116), Snail (#3879), Slug (#9585), FAK (#3285), p-FAK (#3283), AKT (#9272), p-AKT (#4060), Cdc42 (#2466), and GAPDH (#2118), as well as peroxidase-conjugated secondary antibodies were obtained from Cell Signaling Technology, Inc. (Danvers, MA, USA). In addition, radioimmunoprecipitation assay (RIPA) buffer was also obtained from Cell Signaling Technology, Inc. (Danvers, MA, USA).

### 4.4. Cell Viability Assay

Cell viability was assessed using the MTT assay in order to determine cycloartocarpin-mediated cytotoxicity. Cells were seeded at the density of 1 × 10^4^ cells/well in 96-well plates. Following treatment, the cells were then incubated in MTT solution for 4 h at 37 °C. A microplate reader was used to measure the intensity at 570 nm. In addition, the IC_50_ value determination was performed using GraphPad Prism 5 software.

### 4.5. Nuclear Staining Assay

Hoechst33342 and Propidium iodide (PI) co-staining were used to investigate the mode of cell death. Human lung cancer cells were seeded at the density of 1 × 10^4^ cells/well in 96-well plates. After treatment, cells were stained with 10 μM of Hoechst 33,342 and 5 g/mL PI in PBS for 30 min. Cells were imaged by fluorescence microscopy (Olympus DP70, Melville, NY, USA).

### 4.6. Anchorage-Independent Growth Assay

Cells were seeded in 96-well plates at the density of 1 × 10^4^ cells/well and were pre-treated with 0–20 μM cycloartocarpin for 24 h; then, they were assigned to an anchorage-independent growth assay, as previously described [37], for 3 weeks. To avoid dryness, the complete medium was applied every two days. A phase-contrast microscope (Nikon ECLIPSE Ts2, Tokyo, Japan) was used to count and take pictures of the colonies. The size and number of colonies were determined and compared to those of control cells.

### 4.7. Migration Assay

Cells were seeded at the density of 2 × 10^4^ cells/well in 96-well plates for 24 h. Next, a wound was made by creating a line on the cell monolayer with 20 μL pipette tip, and cells were incubated with a non-toxic concentration of cycloartocarpin (0–20 μM) at 37 ◦C for 24 and 48 h. Cells were imaged at indicated time points of 0, 24, 48 h, and the wound space was measured by Image J software (NIH, Bethesda, MD, USA). The percentage of the change in the wound space was analyzed as follows: Change in the wound area (%) = (average area at time (0–24 h, 48 h)/average area at time 0 h) × 100.

### 4.8. Western Blot Analysis

Human lung cancer cells were seeded at density 3 × 10^5^ in a six-well plate for 24 h. After cycloartocarpin treatments, the cells were washed with cold PBS and incubated with a protease inhibitor cocktail (Roche Molecular Biochemicals, Indianapolis, IN, USA) and RIPA lysis buffer. Cell lysates were collected, and the protein content was denatured and loaded onto SDS-polyacrylamide gels. The proteins after separation were transferred onto PVDF or nitrocellulose membranes (Bio-Rad). The membranes were soaked in 5% non-fat dry milk for 1 h and consequently incubated with primary antibody at 4 °C overnight. Then, the membranes were incubated with horseradish peroxidase (HRP)-conjugated anti-rabbit or anti-mouse IgG secondary antibodies for 2 h. Chemiluminescence (Supersignal West Pico; Pierce, Rockford, IL, USA) was then used to find the immune complexes. The expression levels of each protein were quantified using ImageJ software (NIH, Bethesda, MD, USA).

### 4.9. Statistical Analysis

All results are represented as the mean ± SEM derived from at least 3 independent experiments. The statistical analysis was performed using one-way analysis of variance (ANOVA). The statistical significance was considered at *p* < 0.05 (*) and *p* < 0.01 (**). In all experiments, graphs were created using GraphPad Prism 5 (GraphPad Software, San Diego, CA, USA).

## 5. Conclusions

In summary, we demonstrated here for the first time how cycloartocarpin, extracted from *A. gomezianus*, could potentially inhibit cancer cell migration, EMT, and anchorage-independent growth. We also determined the mechanism of action. According to the findings, cycloartocarpin suppressed EMT and FAK/AKT/Cdc42 signaling, which are crucial regulators of migration and metastasis in lung cancer cells (Figure 6). The obtained information could facilitate the development of cycloartocarpin as an antimetastatic agent for lung cancer treatment.

## Figures and Tables

**Figure 1 molecules-27-08121-f001:**
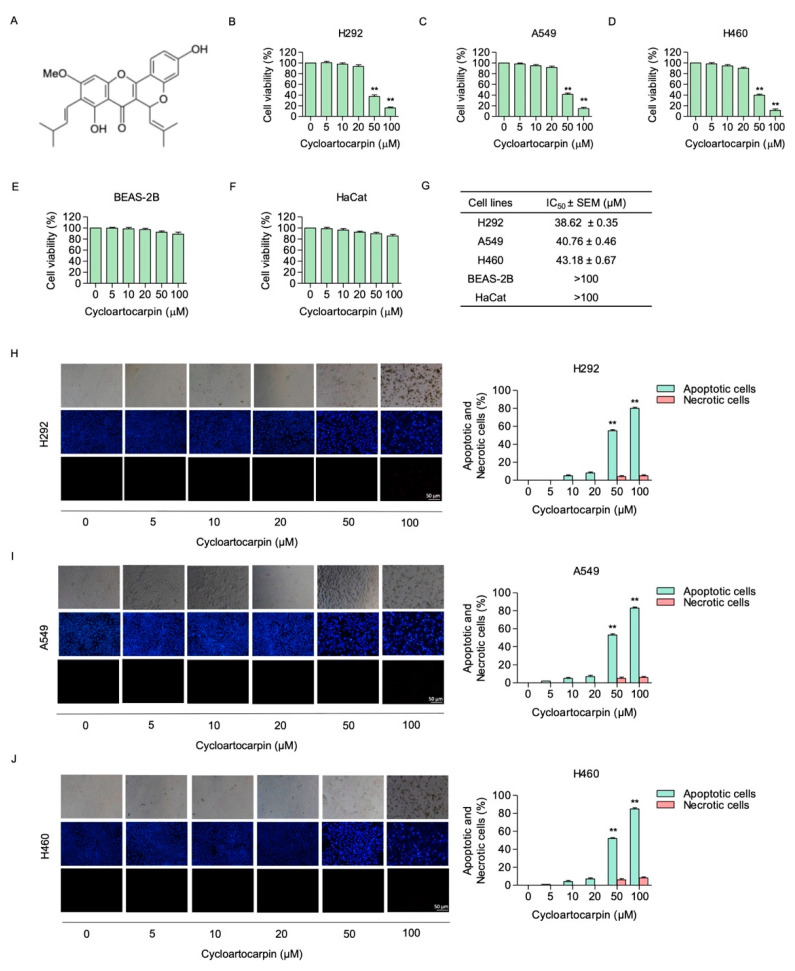
Cytotoxic effect of cycloartocarpin on human non-small-cell lung cancer cells. (**A**) Chemical structure of cycloartocarpin. (**B**–**F**) The cells were treated with cycloartocarpin (0–100 µM) for 24 h. The cell viability was assessed by MTT assay. (**G**) The IC_50_ in all cells was calculated for each cell type using GraphPad Prism 5 software (San Diego, CA, USA). (**H**–**J**) Hoechst 33342/PI staining was used to identify apoptotic and necrotic cells. Images were identified by using a fluorescence microscope. Apoptotic and necrotic cells (%) in cycloartocarpin-treated cells were analyzed. The data are presented as the mean ± SEM (*n* = 3) (** *p* < 0.01, compared with the untreated control).

**Figure 2 molecules-27-08121-f002:**
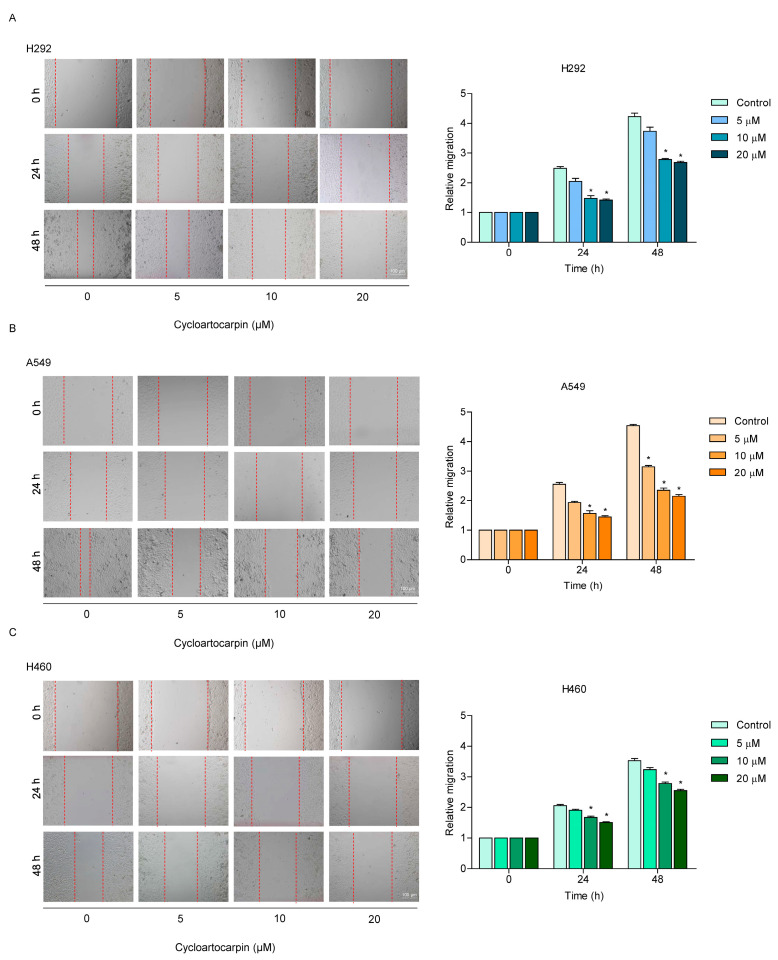
Cycloartocarpin inhibits cell migration in H292, A549, and H460 cells. (**A**–**C**) A wound healing assay was conducted for the migration assay. Wound space was generated, and the cells were treated with 0–20 μM of cycloartocarpin for 24 and 48 h. The data are presented as the mean ± SEM (*n* = 3) (* *p* < 0.05, compared with the untreated control).

**Figure 3 molecules-27-08121-f003:**
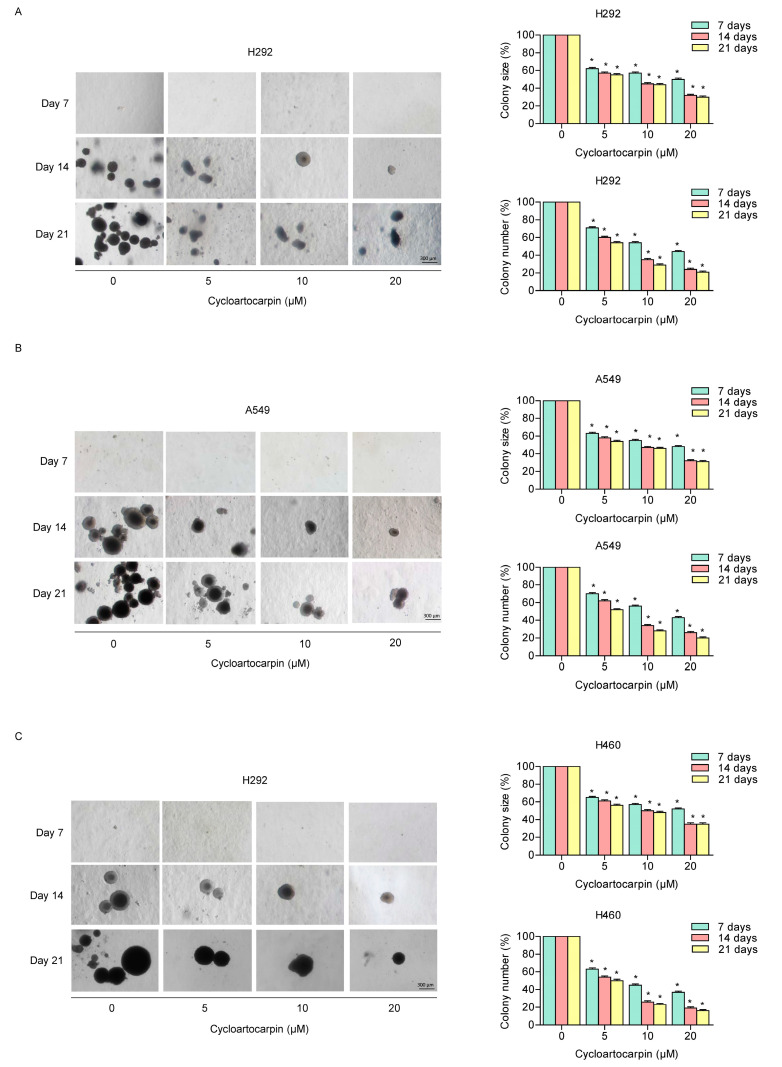
Cycloartocarpin attenuates anchorage-independent growth of human lung cancer cells. (**A**–**C**) H292, A549, and H460 cells were pre-treated with 0–20 μM cycloartocarpin for 24 h. Then, cells were assigned to an anchorage-independent growth assay for 3 weeks. The percentage of size and number were investigated and calculated. The data are presented as the mean ± SEM (*n* = 3) (* *p* < 0.05, compared with the untreated control).

**Figure 4 molecules-27-08121-f004:**
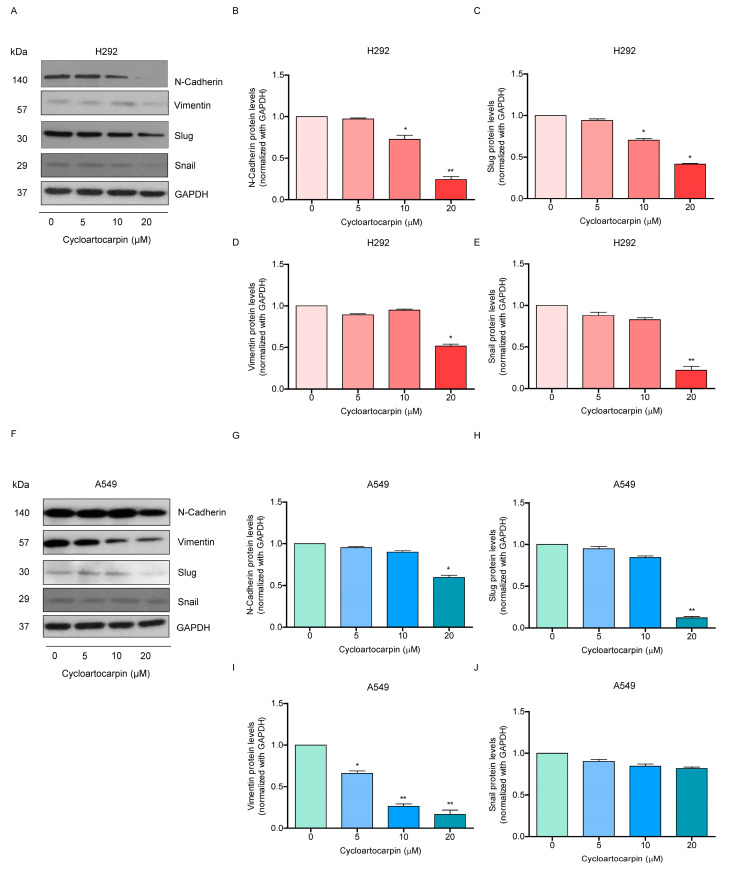
Cycloartocarpin suppresses epithelial-to-mesenchymal transition (EMT) in H292 and A549 cells. (**A**–**E**) H292 cells were treated with cycloartocarpin (0–20 μM) for 24 h. Then, the expression levels of N-cadherin, vimentin, snail, and slug were determined by Western blotting. (**F**–**J**) A549 cells were treated with cycloartocarpin (0–20 μM), and the expression levels of N-cadherin, vimentin, snail, and slug were investigated by Western blotting. To ensure that the protein samples were loaded equally, the blots were repeated using GAPDH. The relative protein levels were estimated by densitometry. The data are presented as the mean ± SEM (*n* = 3) (* *p* < 0.05, ** *p* < 0.01, compared with the untreated control).

**Figure 5 molecules-27-08121-f005:**
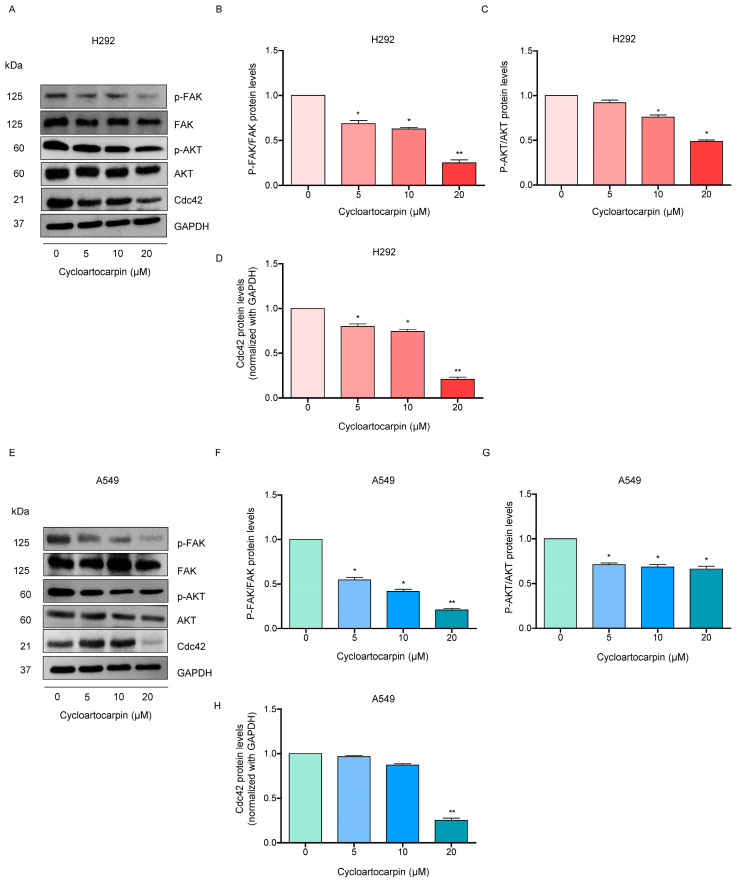
Effect of cycloartocarpin on epithelial-to-mesenchymal transition (EMT) via inhibiting FAK/AKT signaling pathway. (**A**–**D**) H292 and (**E**–**H**) A549 cells were treated with cycloartocarpin (0–20 μM) for 24 h. The protein levels of FAK, phosphorylated FAK (Tyr397), AKT, phosphorylated AKT (Ser473), and Cdc42 were investigated using Western blot analysis. To ensure that the protein samples were loaded equally, the blots were repeated using GAPDH. The relative protein levels were estimated by densitometry. The data are presented as the mean ± SEM (*n* = 3) (* *p* < 0.05, ** *p* < 0.01, compared with the untreated control).

**Figure 6 molecules-27-08121-f006:**
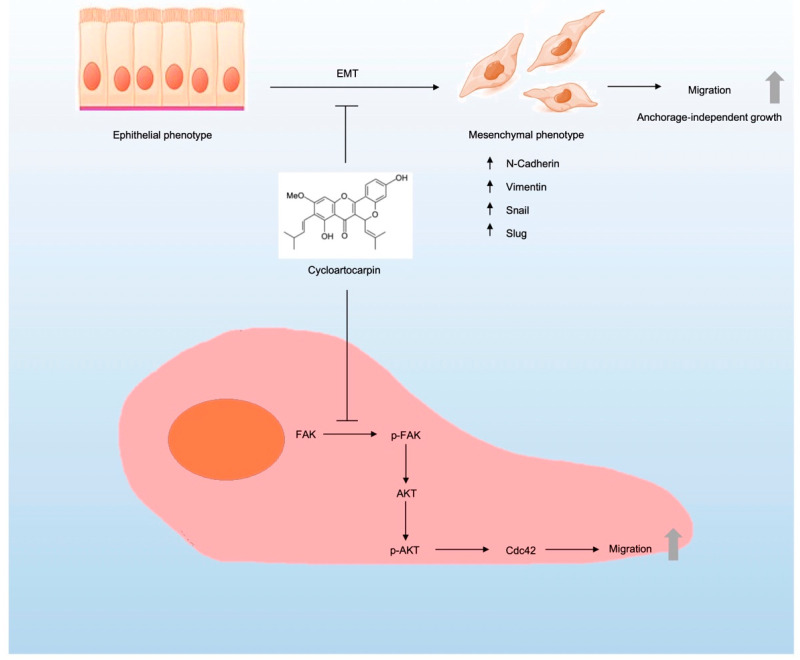
Schematic display of the underlying mechanism of cycloartocarpin in the inhibition of cell migration and EMT process in lung cancer.

## Data Availability

Data are contained within the article.

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
