# Peer review of "Cycloartocarpin Inhibits Migration through the Suppression of Epithelial-to-Mesenchymal Transition and FAK/AKT Signaling in Non-Small-Cell Lung Cancer Cells"

_molecules, 2022, doi:10.3390/molecules27238121_

Round 1
Reviewer 1 Report
In general, “Cycloartocarpin Inhibits Migration through the Suppression of Epithelial to Mesenchymal Transition and FAK/AKT Signaling in Non-Small Cell Lung Cancer Cells” written by Pithi Chanvorachote et al. is a solid work on the investigation of the ability of flavonoid drug cycloartocarpin to prevent lung cancer metastasizing. The set of instruments is good to prove the main concept. Unfortunately, the work lacks important controls to be accepted in its current state. I would recommend this manuscript for publication only after answering the following questions.
1. The work requires at least one cell line which is not of lung cancer type. In the best scenario, it should be a healthy cell line as a control in the assays.
2. There is no explanation of how nuclear staining and fluorescence microscopy helped to count the proportions of healthy/necrotic/apoptotic cells. An explanation should be added or a supporting experiment with the use of Flow Cytometry should be added.
Reviewer 2 Report
In their manuscript (No.: molecules-2028137) authors have described the experimental results of cycloartocarpin effect on the metastasis of non-small cell lung cancer in in vitro conditions. The manuscript shows some interesting results. The results of each step of the study have been well documented. On the basis of the findings obtained, the authors suggested that cycloartocarpin suppress migration and EMT in lung cancer via FAK/AKT/Cdc42 signaling pathway.
Special comments:
1. Why in their experiments the authors used only non-toxic doses of cycloartocarpin? If the IC50 of cycloartocarpin has been determined, why has the effect of the compound at this concentration not been investigated?
2. How the IC50 values for examined drug were determined? This information should be added to the Material and methods section.
3. The material and methods should be more detailed. How did the cycloartocarpin solution prepared? What solvent to prepare stock solution was used? Under what conditions was it stored? The authors conducted a series in vitro studies. Please complete how many cells were seeded in each experiment.
Round 2
Reviewer 1 Report
The comments from the authors and corresponding amendments are sufficient to recommend this work for publication.